# Assessment of *Chrysoperla comanche* (Banks) and *Chrysoperla externa* (Hagen) as Biological Control Agents of *Frankliniella occidentalis* (Pergande) (Thysanoptera: Thripidae) on Tomato (*Solanum lycopersicum*) under Glasshouse Conditions

**DOI:** 10.3390/insects11020087

**Published:** 2020-01-29

**Authors:** Héctor Manuel Luna-Espino, Alfredo Jiménez-Pérez, Víctor Rogelio Castrejón-Gómez

**Affiliations:** Centro de Desarrollo de Productos Bióticos del Instituto Politécnico Nacional, calle Ceprobi No.8, San Isidro, Yautepec, 62739 Morelos, Mexico; hmlunaespino@gmail.com (H.M.L.-E.); aljimenez@ipn.mx (A.J.-P.)

**Keywords:** biological control agent, larvae instars, predator, prey

## Abstract

We tested the predatory capacity of newly-hatched or newly-molted *Chrysoperla comanche* (Banks) and *Chrysoperla externa* (Hagen) larvae after a 24 h fasting period on adults of *Frankliniella occidentalis* (Pergande) that were feeding on tomato plants (at vegetative and blooming stage) under glasshouse conditions. We also recorded fruit damage by the thrips. Both *Chysoperla* spp. depredated a similar number of *F. occidentalis* (thrips) adults regardless of the phenological stage of the plant. Second and third instar larvae of both species consumed significantly more thrips than first instar during plant blooming, however when the plant was at vegetative stage, all larval stages of both species predated a similar number of thrips. A significantly lower fruit damage percentage was recorded at the blooming plant when *C. comanche* larvae were in the experimental cage, however the presence of second and third instar of both species significantly reduced the fruit damage. No foliar damage was recorded. As far as we know, this is the first assessment of the predatory capacity of *C. comanche* and *C. externa* on thrips feeding on tomato under glasshouse conditions.

## 1. Introduction

Mexico is the 10th largest producer of tomato in the world, with 4,047,171 t of fruit [1,2] worth 1666 US million dollars in the USA. However, tomato production is severely affected by diseases and pests, among them the western flower thrips *Frankliniella occidentalis* (Pergande) (Thysanoptera: Thripidae). This insect is a major cosmopolitan pest causing direct and indirect damage to glasshouse-grown horticultural crops [3]. It is the second most important tomato pest as it may damage leaves, flowers and fruits, and transmits the tomato spotted wilt virus and the impatiens necrotic spot virus [4,5,6]. Most farmers control this pest with synthetic insecticides and resistance to different insecticide groups has been declared in different parts of the world [7,8,9,10]. The use of pesticides increases cost production, pollutes the environment and affects human health [11], thus new regulations are in place limiting international and regional trade [12].

The use of pathogens, parasitoids and predators as biological control agents has shown to be successful at glasshouse conditions and its impact have increased worldwide [13]. Phytoseidae mites and Anthocoridae and Miridae bugs have been used widely to control thrips [13,14,15]. The potential of lacewings for controlling thrips or other pests at lab or glasshouse conditions have been assessed as well. For example, *Chrysoperla externa* (Hagen) predating on the thrips *F. occidentalis* [16], *Neohydatothrips signifer* Priesner (Thysanoptera: Thripidae) [17] and *Enneothrips flavens* Moulton (Thysanoptera: Thripidae) [18], the aphids *Schizaphis graminum* (Rondani) [19] and *Myzus persicae* (Sulzer) (Hemiptera: Aphididae) [20], the whitefly *Trialeurodes vaporariorum* Westwood (Homoptera: Aleyrodidae) [21], the psyllid *Diaphorina citri* Kuwayama (Hemiptera: Psyllidae) [22] and the spotted wing drosophila, *Drosophila suzukii* Matsumura (Diptera: Drosophilidae) [23].

*Chrysoperla carnea* (Stephens) successfully controlled aphids on peppers, cucumber, eggplant and lettuce [24]. Third instar larvae of *C. carnea* preferred the lettuce aphids, *Nasonovia ribisnigri* (Mosley) (Hemiptera: Aphididae) as prey over the thrips *F. occidentalis* under laboratory conditions [25]. Inundative releases of *Chrysoperla rufilabris* (Burmeister) reduced the population of *Bemisia tabaci* feeding on *Hibiscus rosa-sinensis* at glasshouse conditions so efficiently that maintained all plants in a marketable condition [26]. Recently, it was suggested that *Chrysopa pallens* (Rambur) could be a good biological control agent of *F. occidentalis* colonizing cucumber at glasshouse conditions [27]. Our paper reports the potential of *Chrysoperla comanche* (Banks) and *C. externa* (Hagen) as a biological control agent of *F. occidentalis*, feeding on tomato under glasshouse conditions.

## 2. Materials and Methods

### 2.1. Plant Production and Maintenance

Tomato plants used in this research were grown from seeds germinated using a mixture of perlite (20%), vermiculite (20%) and peat moss (60%) resulting in a plant substrate with 85% porosity and a 5.8 ± 0.3 pH. When the plants had 4–5 true leaves, they were transferred to 10 L black polyethylene bags filled with the same substrate plus 20% of tezontle to improve water drainage and substrate consistency. Plants were kept in a glasshouse at 28–36 °C, 32%–52% RH and a 12:12 light regime.

For bioassays we used the local commercial tomato variety, the Rio Grande^®^ tomato (*Solanum lycopersicum* L.) and for breeding thrips we used the Verde Paris^®^ cucumber (*Cucumis sativus* L.). Normal tomato and cucumber culture practices were implemented at the glasshouse. 

### 2.2. The Insects

(a) Frankliniella occidentalis colony.

Young cucumber plants were kept out of our glasshouse in 10 L bags to be colonized by thrips. Once the plants were infested by thrips, they were introduced into a glasshouse for protection and maintenance. Thrips were morphologically identified as *Frankliniella occidentalis* (Pergande) [28]. 

(b) *Chrysoperla comanche* and *Chrysoperla externa*.

Adult males and females of both species of *Chrysoperla* were collected from avocado trees at Tetela del Volcán, Morelos, Mexico (18°53′35″N 98°43′47″O). Morphological identification of the insects collected at Tetela del Volcán produced *Chrysoperla comanche* (Banks) and *Chrysoperla externa* (Hagen) [29]. A lab colony (26 ± 5 °C, 35 ± 20% RH and a 12:12 L: D photoperiod) was established for each species using plastic containers (23.5 cm long, 5 cm high and 15 cm wide) and adults were fed the diet reported by Vogt et al. [30]. The mouth of the plastic container was covered with a clean cloth where females laid their eggs. The cloth with the eggs was collected daily and incubated in the above-described container at the lab colony conditions. On emergence, eggs of *Sitotroga cerealella* Olivier (Lepidoptera: Gelechiidae) were provided as food for the larvae until the *Chrysoperla* spp. turned into pupae [31,32].

### 2.3. Bioassays

We tested the predatory capacity of the first, second and third instar larvae of both *Chrysoperla* species under two phenological stages of the plant; vegetative growth and blossom. Also, we evaluated thrips’ damage to fruits 

(a) The predators.

First, second and third instar larvae of *Chrysoperla comanche* and *C. carnea* were obtained from the corresponding colony 24 h before the experiment and no food was provided. Larvae were kept in plastic containers at the lab colony conditions. Larvae were observed under a stereoscope to ensure they were healthy and fit prior to use. 

(b) The prey.

Adult thrips were collected from the colony and keep at the glasshouse using an aspirator. Insects were observed under a stereoscope to ensure only healthy and fit insects were used in the experiment.

(c) The host.

Vegetative growth: Tomato plants were 30–40 cm tall and were grown as mentioned before. Plants were allocated randomly into the groups to reduce any effect related to plant size. 

Blossom: This experiment was carried out when the plants had three bouquets. 

(d) The prey, the predator and the plant.

A total of 40 or 90 thrips were released on a tomato plant (vegetative growth or blossom, respectively) into each experimental unit, waiting 5 min before introducing the predator. A total of nine larvae of first, or second or third instar larvae of *C. externa* or *C. comanche* were allocated to each experimental unit (three larvae of the same instar for each plant and nine plants per treatment). Tomato plants with 40 or 90 thrips without predators were used as control. All predators and preys were recovered 24 h later and observed under the stereoscope to differentiate those thrips consumed by the predator. Predated thrips present an empty and squeezed body. In addition, we observed the first damage caused by thrips in small fruits from 15 to 21 days and two and a half months later at harvest time, we counted the number of fruits damaged by thrips (punctured fruits or with scars) in each of the evaluated plants.

Three blocks were used for each species. Each block had four experimental units (three plants each) (*N* = 36 plants for each *Chrysopa* species including its control). Blocks and experimental units were covered with anti-aphid mesh to prevent insect migration. All experiments were conducted under the thrips’ colony conditions. 

### 2.4. Statistical Analysis

The number of *Frankliniella occidentalis* consumed by *Chrysoperla comanche* and *Chrysoperla externa* larvae at the vegetative and blossom stage of the tomato plant were analyzed by a χ^2^ tests of independence. The number of thrips consumed by each larval stage at both phenological plant stages was analyzed by a Kruskal–Wallis test followed by a Holm–Sidak test as data failed the Shapiro–Wilk test and Levene’s mean test for normality and homoscedasticity, respectively. A two-way ANOVA (species and instars as factors) followed by a Holm–Sidak test was used to compare the percentage of damaged fruits prior to arcsine transformation of the original percentage data to meet the assumptions of normality and homoscedasticity, however, original data is reported. All statistical analysis was carried out in SigmaPlot 12.5 at a 0.05 rejection probability. 

## 3. Results

### Consumption of Frankliniella occidentalis by First, Second and Third Instar of Chrysoperla comanche and Chrysoperla Externa on Tomato Plants

Both species predated a similar number of thrips regardless of the phenological stage of the plant (for vegetative χ^2^ = 0.632, df = 2, *p* > 0.05 and for blossom stage χ^2^ = 1.686, df = 2, *p* > 0.05) (Table 1). *Chrysoperla comanche* and *C. externa* larvae consumed 21.6% and 25%, respectively, of the available prey at the vegetative stage; similarly prey consumption was 31.8% and 36.6%, respectively, of the available prey at blossom (Table 1). The number of thrips consumed during the vegetative stage of the plant by the different instars and species were similar (Overall, Q_1_ = 2, Median = 3, Q_3_ = 4; H = 6.632, df = 5, *p* = 0.2). However, at blossom, second and third larvae of both *Chrysoperla* species consumed more thrips than the first instars ones (H = 39.56, df = 5, *p* > 0.001) (Figure 1). Second and third instar larvae of both *Chrysoperla* species accounted for 73.2% and 90.8% of the kills at the vegetative and blossom stage, respectively (Table 1). 

Plants infested with thrips and protected with *C. externa* presented significantly less fruits damaged (mean = 44.1 ± SEM = 0.04%) than those protected by *C. comanche* larvae (mean 56.4 ± SEM = 0.04%) (F = 4.67, df = 1,64, *p* = 0.03). Similarly, the plants protected with second and third instars presented less fruits damaged than those protected with first instars larvae and the control group (F = 11.88, df = 3,64, *p* < 0.001). The interaction species x instars was not significant (F = 0.028, df = 3,64, *p* > 0.05) (Figure 2). The fruit in the control group was 1.3, 2.5 and 3 times more damaged than when first, second and third larvae were in the tomato plant. 

## 4. Discussion

The presence of both *Chrysoperla* spp. larvae reduces the thrips population. This reduction is more pronounced at blooming than at the vegetative stage and in any case, this value is not greater than 36.6% in a 24 h period. Our data indicate that in both species, second and third instar larvae consume more preys than these of the first instar. Similar results have been reported by several authors but with different prey, plant or predator combinations and at lab conditions. For example, the third instar of *C. externa* consumed more thrips (*F. occidentalis*) [16] and *Neohydatothrips signifier* [17] than the second and first instar. The presence of newly hatched *C. externa* larvae proved to significantly reduce *E. flavens* populations on peanut plants (*Arachis hypogaea* L.; Fabales: Fabaceae) under greenhouse conditions substantiating its potential as a biological control agent [18]. The third instar of *C. carnea* was the most voracious when feeding on the potato psyllid, *Bactericera cockerelli* (Sulc) [33], while third and second instars of *C. externa* consumed the largest number of whiteflies on tomato leaves at lab conditions [34]. Third instar *C. carnea* larvae readily preyed upon both thrips and aphids, showing preference for aphids in laboratory conditions [25]. First instar larvae of both species consumed fewer thrips at the vegetative or blossom stage than the other two instars. This reduction on prey consumption could be related to three different facts: a) the small size and reduced mobility of the predatory larvae, b) thrips tend to jump or to have short flies in the presence of the predator, making catching a prey a difficult task [16] and c) the glandular trichomes of the tomato plant provides shelter and cover, making it more difficult to capture preys as was found out when *C. externa* larvae predated on the whitefly *Bemisia tabaci* [35].

The larger number of preys consumed by the predators during tomato blossom could be the result of three aspects: (a) thrips tend to congregate at or nearby the flowers, aiming for tender plant tissue, (b) thrips are attracted to blue and yellow colors so they congregated on the yellow flowers, as was reported for *F. occidentalis* being more abundant on pepper, cucumber and zucchini flowers [36], and (c) a larger number of preys available in the experimental unit than at the vegetative stage. This could be partially true, as second and third larvae consumed more preys than first instar ones (Table 1), but first instars larvae consumed similar number of preys, regardless of the large number of preys available at the blossom stage. 

Experimental conditions greatly affect the predatory capacity of *Chrysoperla* spp. Many studies at lab conditions use a leaf of a host or a leaf disc inside a container. Under this circumstances, a single first, second and third instar larvae of *C. externa* consume an average of 27.2, 26.1 and 34.4 larvae I and II of *N. signifier* on a maracuya leaf daily [17], while when preying on adults of *F. schultzei*, their average daily consumption was 3.3, 9.7 and 10.2 adults, respectively, per day in the presence of a disc of lettuce [37]. A daily average consumption of 3.5, 3.8 and 5.5, respectively, of *F. occidentalis* adults was reported when no host is present in the experimental arena [16]. Our results indicate that there is a clear relationship between the number of thrips and its larval instar and the percentage of fruit damaged. A low population of the thrips *Heliothrips haemorrhoidalis* Bouché, *F. invasor* Sakimura, *Scirtothrips perseae* Nakuhara, and *S. hectorgonzalazi* Johansen & Mujica has been correlated with a minor number of mango fruit damaged [38]. Minor damage to grapes was reported when the population of *F. occidentalis* decreased [39], while an increase in the number of the thrips *N. signifer* in the maracuya flower reduced the fruit yield significantly [40]. 

Both *Chrysoperla* spp. are good biological control agents and are feasible to implement an augmentative strategy for the control of *F. occidentalis* on tomato. Simultaneous releases of both lacewing species could control *F. occidentalis* as has been demonstrated for other predators. For example, when the mites *Amblyseius cucumeris* Oudemans, *Hypoaspis aculeifer* Canestrini and the bug *Orius insidious* Say were released simultaneously they achieved good control of *F. occidentalis* (Pergande) [15]. The same level of control of *F. occidentalis* (Thysanoptera: Thripidae) was achieved when the mite *Amblyseius* (*Typhlodromips*) *swirskii* (Athias-Henriot) (Acari:Phytoseiidae) and the bug *Orius insidiosus* (Hemiptera: Anthocoridae) were released on *Rosa hybrida* L. cv. Tropicana at lab conditions [14]. However, in future work it will be necessary to investigate whether the release of both *Chrysoperla* spp. is compatible or a predation between predators can occur (intraguild predation), as has happened in other predators.

Lacewing populations in the glasshouses can be established and maintained by introducing plants that provide pollen and nectar for the adults and supplying *Sitotroga cerealella* (Olivier) eggs for the larvae when the thrips population is scarce, establishing an augmentative or conservation biological control as has been proposed in other studies [18,25]. This strategy has been implemented to maintain a population of predatory mites, by introducing plants to supply pollen and lab reared mites as prey for the predatory mites when the thrips population in the glasshouse is low or totally absent [13]. Lacewings can be used in an IPM (Integrated Plant Management) system for *F. occidentalis* [41] including the use of colored traps lured with sex [42,43] or aggregation pheromones [44,45,46].

Releasing a large number of *Chrysoperla* spp. eggs six days before blossom can reduce fruit damage under glasshouse conditions. However, there is no information on some relevant aspects of the lacewings and thrips species. At the moment we don’t know how many and which *Chrysoperla* larvae must be released by tomato plant in a glasshouse to achieve good control. Releasing both species of *Chrysoperla* at the same place and time would increase the efficiency of the biological control or prey competition (interspecific competition), and how this strategy may affect the functional response and population growth of both *Chrysoperla* species to achieve a predictive, efficient and cost-effective particular biocontrol model for these predator and thrips has been suggested by Plouvier and Wajnberg [47]. Therefore, more research is required on the life-story parameters of the prey and the predator(s) and their relationship with the host plant. This is the first report of the predatory capacity of *C. externa* or *C. comanche* on tomato under glasshouse conditions.

## 5. Conclusions

The results of this study showed that *Chrysoperla comanche* and *Chrysoperla externa* can be used to control *Frankliniella occidentalis* in tomato grown in a greenhouse, regardless of the phenological state of the plant. When the plant was in its vegetative stage, the number of thrips consumed was similar regardless of instars and species, but in the blossom stage the second and third instar larvae of both *Chrysoperla* species consumed a greater number of thrips than the first instar ones. Plants in which *C. externa* larvae were released showed significantly less fruit damaged by thrips than in plants with *C. comanche* larvae. It was also demonstrated that plants with second and third instars larvae, presented significantly less damaged fruits than those with first instar larvae and the control group.

## Figures and Tables

**Figure 1 insects-11-00087-f001:**
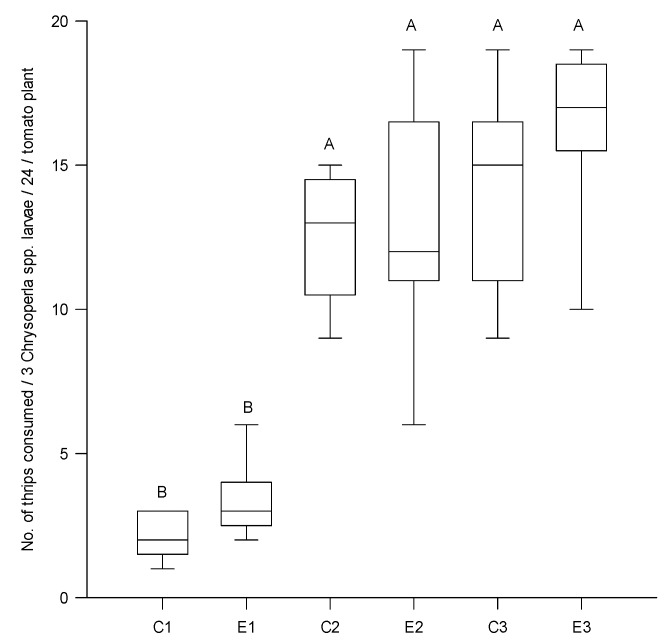
Number of thrips consumed by *C. externa* and *C. comanche* first, second and third instar larvae on tomato plants during blossom. Data are Q_1_ < median < Q_3_. KW, H = 39.56, df = 5, *p* > 0.001. Bars topped by the same letter are not different (Holm–Sidak, *p* > 0.05). C = *C. comanche*, E = *C. externa*. First, second and third instar are represented by 1, 2 and 3, respectively.

**Figure 2 insects-11-00087-f002:**
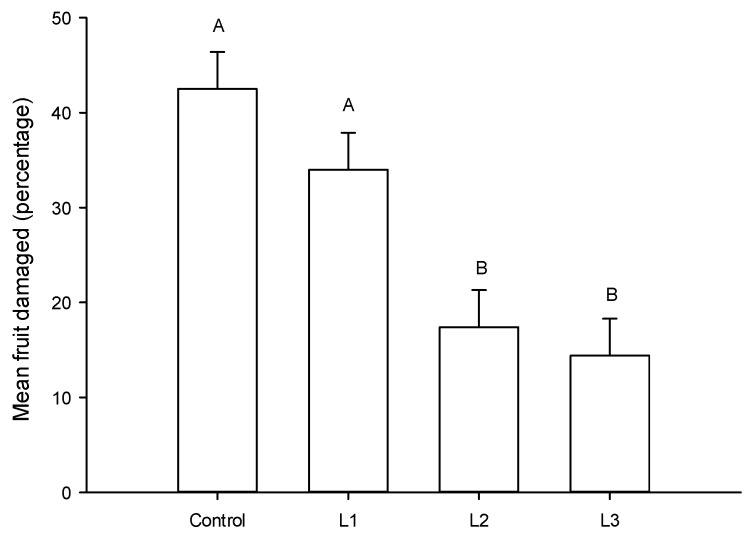
Mean (±SEM) percentage of tomato fruit damaged by *Frankliniella occidentalis* when first (L1), second (L2) and third (L3) instar larvae of *Chrysoperla comanche* or *Chrysoperla externa* were in the tomato plant. Original data are presented. Error bars are SEM. Bars followed by the same letter are not significantly different (Holm–Sidak, *p* > 0.05).

**Table 1 insects-11-00087-t001:** Adults of *Frankiniella occidentalis* consumed in a 24 h period by first, second and third instar larvae *of Chrysoperla comanche* and *Chrysoperla externa* on tomato plants at vegetative and blooming stage under glasshouse conditions. Letters in bold are the sum of the adults thrips consumed by each *Chrysoperla* species.

Species	Vegetative Stage X^2^ = 0.632, df = 2, *p* > 0.05				Blooming Stage X^2^ = 1.686, df = 2, *p* > 0.05				Total
	L_1_	L_2_	L_3_	Total	L_1_	L_2_	L_3_	Total	
***C. comanche***	19	29	30	78	20	113	125	258	336
*C. externa*	26	34	30	90	31	118	148	297	387
Total	45	63	60	168	51	231	273	555	723

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
