# Peer review of "Assessment of Chrysoperla comanche (Banks) and Chrysoperla externa (Hagen) as Biological Control Agents of Frankliniella occidentalis (Pergande) (Thysanoptera: Thripidae) on Tomato (Solanum lycopersicum) under Glasshouse Conditions"

_insects, 2020, doi:10.3390/insects11020087_

Round 1

Reviewer 1 Report

All the indicated changes have been addressed. I have no further comment on the manuscript, which I recommend to be accepted

Author Response

Reviewer 1 no  longer  had comments or  corrections and recommended  accepting the manuscript

Reviewer 2 Report

This manuscript titled ‘Assessment of Chrysoperla comanche (Banks) and Chrysoperla externa (Hagen) as biological control agents of Frankliniella occidentalis (Pergande) (Thysanoptera: Thripidae) on tomato (Solanum lycopersicum) under glasshouse conditions’ has been revised and the authors made significant changes to the manuscript.

There are still minor changes to be made (mentioned below):

The introduction section has the objective but is still missing hypotheses of the study. Mention or explain what morphological key was used for identification of thrips and the authors deleted the text related to molecular testing. More details in the methods are required. In statistical analyses, the authors completely removed the text related to G-tests. Please include and mention why G-tests were used! Correct the spelling - arcsine.

Line specific comments below:

Line 15 – correct the spelling to thrips! Line 21 – change to significantly reduced instead of reduced significantly Line 23 – on thrips ‘feeding’ on tomato Line 27 – remove spaces between the digits – 4,047,171 and add space before t Line 29 – among them ‘is’ the Line 34 – to different ‘insecticide’ groups Line 50 – Bemisia not Bemicia Lines 59, 62, 75 – add single space before % or oC Line 100 – thrips were ‘released’ Line 139 – Figure 1, correct the spelling ‘thrips’ in the y-axis label Line 175 - Bemisia not Bemicia Lines 157, 196, 199, 211, 221 – ‘thrips’ not thrip!!

Author Response

Reviewer 2

The introduction section has the objective but is still missing hypotheses of the study. Mention or explain what morphological key was used for identification of thrips and the authors deleted the text related to molecular testing. More details in the methods are required. In statistical analyses, the authors completely removed the text related to G-tests. Please include and mention why G-tests were used! Correct the spelling - arcsine.

We consider that the hypothesis is implicit in the last paragraph of the introduction section, and reviewing other articles in Insects, we did not find that the hypothesis is written as such. The references of the taxonomic keys to identify the thrips and the Chrysoperla species were annexed in the manuscript, L75 and L80 respectively:

28. Hoddle, M. S.; Mound, L.A.; Paris, D. L. Thrips of California. CBIT. Publishing. Queensland. 2012. Disponible:http://Keys.lucidcentral.Org/keys/v3/thrips_of_california/Thrips_of_california.hml. (Consultado en enero de 2017).

29. Brooks, S. J.; Barnard, P. C. The green lacewings of the world: a generic review (Neuroptera: Chrysopidae). Bull. Br. Mus. Nat. Hist. Entomol. 1990, 59: 117-286.

Indeed, we delete the text of molecular identification because we already explained that this information is the reason for another publication. We had included more details in the Methodology section. See L109 and L 113-115. And regarding the statistical analysis ...Reviewer No. 3 called for the replacement of the G-test by the χ2 test. See previous Reviewer 3 report. We initially used a G-test because G-values are additive and can be used for elaborated statistical designs. The G-test is a likelihood test while is χ2 a score test. Low frequency values invalidated the χ2 test but, having large frequency values in a table, the sensitivity to low frequency values of the χ2 is not a problem for analyzing our data. Both tests give similar results. Arcsin was changed by arcsine.

Line specific comments below

L15 =  thrips; was corrected throughout the document

L21 = reduced significantly; was changed by significantly reduced

L23 = on thrips on tomato; was changed by on thrips feeding on tomato

L27= remove spaces between the digits – 4,047,171 and add space before t ; was made

L29= among them the;  was  changed  by among them is the

L34= insecticides; was changed  by insecticide

L50= Bemicia; was  changed  by Bemisia

L59, 62,75= add single space before % or °C; the space was added in all cases

 L100= thrips were ‘release’; was  changed  by thrips were released

L139 = Figure 1, correct the spelling ‘thrips’ in the y-axis label; trips were changed by thrips in the y-axis label

 L175= Bemicia; was  changed  by  Bemisia 

L157, 196, 199, 211, 221= ‘thrips’ not thrip!; thrip were changed by thrips

Reviewer 3 Report

The paper has been improved and almost all of the comments have been integrated into the text. The main critical points are: 1) The results of the present paper are highly specific (control of tomato infestation by F. occidentalis by two species of Chrysoperla different by C. carnea) and in my opinion of low interest in people not involved specifically in this topic. 2) Authors (lines 198-200) suggest the introduction of both species to increase the control of pest, but they did not approach, in present paper, problems associated with negative interaction between predators (such as intraguild predation). Therefore this conclusion can not be supported by data in present research.   In addition I add some notes directly in the text. Even if English has been improved respect to previous version and even if I'm not a mother language, I still suggest a check of English.

Author Response

Reviewer 3

Comments and Suggestions for Authors

The paper has been improved and almost all of the comments have been integrated into the text. The main critical points are: 1) The results of the present paper are highly specific (control of tomato infestation by F. occidentalis by two species of Chrysoperla different by C. carnea) and in my opinion of low interest in people not involved specifically in this topic. 2) Authors (lines 198-200) suggest the introduction of both species to increase the control of pest, but they did not approach, in present paper, problems associated with negative interaction between predators (such as intraguild predation). Therefore this conclusion can not be supported by data in present research.   In addition I add some notes directly in the text. Even if English has been improved respect to previous version and even if I'm not a mother language, I still suggest a check of English.

Respectfully, we believe that reviewer 3 did not understand the paragraph of lines L231 to 236, since we wrote that releasing both species of Chrysoperla could increase the biological control of thrips, but in no way are we claiming that they would control thrips. We also deleted what was related to intraguild predation, precisely because our work did not address competition among predators. The notes in the text were taken care of and the english was checked.

L107 = the authors should explain what they mean by damaged: non eatable, inadequate looking. It is also important to understand how they evaluated it: for example for a number of thrips greater than xx?; The following text was added: In addition, we observe the first damage caused by thrips in small fruits from 15 to 21 days and two and a half months later at harvest time, we count the number of fruits damaged by thrips (punctured fruits or with scars) in each of the evaluated plants.

L118 = by which tests? by Shapiro-Wilk Normality test and Levene’s mean test for homoscedasticity. This information has been included in the L117-118. 

139 = Figure 1, correct the spelling ‘thrips’ in the y-axis label

L139 = I suggest reversing the order so that it is the same as in Tab 1 and Fig. 2 (L1, L2, L3).  Done accordingly

L 157 = What mean certain? For example it can be adequate or not for tomato market. It may be significant or insignificant from a statistic point of view.         We decided to delete “at a certain degree”

L158 = statistically significant? The number of preys and size of the plants were different at the vegetative and blooming stage. This condition prevents any statistical comparison.  

L172 = In my opinion the explanations 2 and 3 are still associated with explanation 1. For example "thrips tend to jump" both for L1, L2 and L3 larvae, but I supposed that a greater dimension (L2 and L3) allow to catch the thrips before they flight. Therefore hereagain the ultimate explanation is that L2 and L3 are greater than L1. Less clear is the explanation 3: a lower dimension should allow L1 to introduce in shelter and catch thrips.

This point should be better explained by authors.

With all due respect, we do not agree with Reviewer 3. In explanation 1 we talk about the small size and reduced mobility of the Chrysoperla  larvae. In explanation 2 we talk about the behavior of the thrips to escape by means of jumps or flights. And in explanation 3 we talk about the glandular trichomes of the plant that give shelter and cover to the thrips.

L183 = This is the same as explanation 1 or 2. More congregations in blossom means lower density in vegetative plant.

L183 = We agree with Reviewer 3, the following text was deleted; …and d) predators at the vegetative stage failed to consume more preys due to an increased difficulty to locate the prey as a result of a lower prey density.

L199 = F. occidentalis was included in the text.

L199-202 = The present paper does not try to understand the positive, neutral or negative interaction between the two Chrysoperla species. The presence for example of intraguild predation, as demonstrated in several predators, can reduce instead of increase predatory pressure.

We did not want to address the issue of predation between species because it was not the objective of the present work. However, we include some information about it at the end of paragraph, L205 = However, in future work it will be necessary to investigate whether the release of both Chrysoperla spp. is compatible or a predation between predators can occur (intraguild predation), as has happened in other predators.

L202 = The  text  …Say they achieved good control of F. occidentalis (Pergande) [15].; was modified as …Say were released simultaneously they achieved good control of F. occidentalis (Pergande) [15].

L212 = for details see was  deleted

L221. The text  …as has been suggested [45]. ;  was  modified  as … as has been suggested by Plouvier and Wajnberg [47].

L310 = were italicized

L356 = were italicized

This manuscript is a resubmission of an earlier submission. The following is a list of the peer review reports and author responses from that submission.

Round 1

Reviewer 1 Report

This manuscript provides a perspective on the assessment of two Chrysoperla species as biological control agents of Frankliniella occidentalis on tomato plants under glasshouse conditions.

While the manuscript is comprehensive and well written, I have provided a some suggested edits to improve grammar and spelling throughout the text. After those, I believe the manuscript has merit for publication after minor revision. My detailed suggested edits and comments are provided in the attached pdf file.

Reviewer 2 Report

This manuscript titled ‘Assessment of Chrysoperla comanche (Banks) and Chrysoperla externa (Hagen) as biological control agents of Frankliniella occidentalis (Pergande) (Thysanoptera: Thripidae) on tomato (Solanum lycopersicum) under glasshouse conditions’ provides information about the two parasitoids and their effect on Western flower thrips.

The authors used thrips infested tomato plants and performed bioassays involving various factors. The results of this study mention or conclude that the above two parasitoids can be used to control Western flower thrips feeding on tomato plants.

The title of the manuscript is very interesting; however, it is very poorly written. The authors should be aware that, ‘thrips’ is both singular and plural form (never trips or trip). The abstract section has lot of typos and several scientific names in the paper are not italicized. Its ‘tomato spotted wilt virus’ not wild! The authors should have paid keen attention to these basic details, where thrips and parasitoids are the main characters of the study.

The introduction section is not detailed enough. It requires a strong background and the authors failed to mention the main objective and hypotheses of their study.

The authors nowhere mentioned how many biological replicates were used in the study. They also say that thrips and insects were tested morphologically and molecularly, with no additional details. This part of the materials and methods is important as well and should be included as supplementary files.

In statistical analyses, explain briefly what G-tests are and correct the typos (Eg. Arcsine).

Similar to Figure 1, Figure 2 is also best represented with a box plot instead of a bar graph.

All the sections in the paper need significant additional information to have a strong foundation which will support your results. Current form of the manuscript has a lot of missing information and numerous mistakes which has to be fixed.

Reviewer 3 Report

The paper deal with the potential of Frankliniella occidentalis biological control by Chrysoperla externa and C. comanche on tomato. Although the article reports some interesting data in particular relating to the impact of these two predators, much of the information was still known. For example, predation by C. externa on F. occidentalis had already been documented (Lune et al., 2017 as quoted by the authors). The importance of Chrysoperla (although carnea) in the control of F occidentalis has also been studied in Shrestha and Enkeraard 2013 (not mentioned by the authors). The importance of C. externa in the control of other thrips pecies has been for example documented in Rodrigues et al., 2014 (not mentioned by authors). On the other hand some interesting aspects such as intraguild predation were not addressed by the authors (as underlined in the text: lines 207-212)

The statistical methods used by the authors do not seem adequate and / or are not explained in detail. For example, the G-test is suggested for comparisons with high numbers, a condition not satisfied in the present research. On the other hand, the authors do not indicate why they use a non-parametric test such as the Kruskal Wallis or whether the assumptions for ANOVA are satisfied.

I’m not a mother language but the paper need a check for English: in several points (as reported in the attached file) the meaning of the sentences is not clear. In addition the text needs a careful check as underlined in the attached file.
